# Differences in Kidney Function Estimates Based on Creatinine and/or Cystatin C in Non-Traumatic Amputation Patients and Their Impact on Drug Prescribing

**DOI:** 10.3390/jcm8010089

**Published:** 2019-01-14

**Authors:** Mia Aakjær, Morten B. Houlind, Charlotte Treldal, Mikkel Z. Ankarfeldt, Pia S. Jensen, Ove Andersen, Esben Iversen, Lona L. Christrup, Janne Petersen

**Affiliations:** 1Clinical Research Centre, Copenhagen University Hospital, 2650 Hvidovre, Denmark; morten.baltzer.houlind@regionh.dk (M.B.H.); charlotte.treldal.02@regionh.dk (C.T.); mikkel.zoellner.ankarfeldt@regionh.dk (M.Z.A.); ann.pia.soee.lytken.jensen@regionh.dk (P.S.J.); ove.andersen@regionh.dk (O.A.); esbeniversen1@gmail.com (E.I.); petersen.janne@gmail.com (J.P.); 2Section of Pharmacotherapy, Department of Drug Design and Pharmacology, University of Copenhagen, 2100 Copenhagen, Denmark; llc@sund.ku.dk; 3The Capital Regional Pharmacy, 2730 Herlev, Denmark; 4Center for Clinical Research and Prevention, Copenhagen University Hospital, 2000 Frederiksberg, Denmark; 5Department of Orthopaedic Surgery, Copenhagen University Hospital, 2650 Hvidovre, Denmark; 6Emergency Department, Copenhagen University Hospital, 2650 Hvidovre, Denmark; 7Section of Biostatistics, Department of Public Health, University of Copenhagen, 1014 Copenhagen, Denmark

**Keywords:** Creatinine, cystatin C, glomerular filtration rate, renal insufficiency, amputation, drug therapy, drug dose adjustment, drug dosing, inappropriate prescribing

## Abstract

Accurate kidney function estimates are necessary when prescribing renally-eliminated medications. Our objectives were to investigate how amputation affects estimated glomerular filtration rate (eGFR) and to determine if dosing recommendations differ among different eGFR equations. In a cohort study of non-traumatic amputation patients, eGFR based on creatinine and/or cystatin C were measured before and after amputation. Prescribed, renally-eliminated medications were compared with dosing guidelines in Renbase^®^. Data from 38 patients with a median age of 75 years were analyzed. The median (range) eGFR was 65 (15–103), 38 (13–79), and 48 (13–86) mL/min/1.73 m^2^ before amputation and 80 (22–107), 51 (13–95), and 62 (16–100) mL/min/1.73 m^2^ after amputation for eGFR_Creatinine_, eGFR_CystatinC_, and eGFR_Combined_, respectively (*p* < 0.01). From before to after amputation, eGFR increased on average by 8.5, 6.1, and 7.4 mL/min/1.73 m^2^ for eGFR_Creatinine_, eGFR_CystatinC_, and eGFR_Combined_ (all *p* < 0.01), respectively. At least one renally-eliminated medication was prescribed at a higher dose than recommended in 37.8% of patients using eGFR_CystatinC_, 17.6% using eGFR_Combined_ and 10.8% using eGFR_Creatinine_. In conclusion, amputation affects eGFR regardless of the eGFR equations. The differences among equations would impact prescribing of renally-eliminated medications, particularly when switching from creatinine to cystatin C.

## 1. Introduction

Inappropriate drug prescribing is associated with adverse drug reactions, toxicity, and mortality [1]. Approximately 40% of all drugs or their active metabolites are excreted by the kidneys, posing a challenge for accurate dosing in patients with reduced kidney function [2]. These renally-eliminated medications must be prescribed according to kidney function to optimize drug therapy. Previous observational studies from hospital settings found that prescribing renally-eliminated medications at a dose higher than that recommended by clinical guidelines occurs in up to two-thirds of patients [3,4,5]. Meanwhile, a Swedish study concluded that one-third of all adverse drug reactions leading to hospitalizations were related to improper dose adjustment according to the patient’s kidney function [6].

Patients undergoing non-traumatic lower extremity amputation (hereafter referred to as amputation) are often poly-medicated and suffer from reduced kidney function [7]. Furthermore, these patients have high rates of comorbid cardiovascular disease, diabetes, and inflammation, which can accelerate the loss of kidney function. Typical medications for amputation patients include, renally-eliminated medications, such as opioids, adjuvant analgesics, antithrombotic drugs, and non-steroidal anti-inflammatory drugs [7], which highlight the importance of obtaining an accurate measurement of kidney function for this group of patients.

Exogenous filtration markers such as inulin and iohexol are accurate measures of glomerular filtration rate (GFR), but these “gold standards” are expensive, time-consuming, and not part of standard clinical practice. Instead, GFR is typically estimated with an endogenous biomarker. Creatinine is most frequently used worldwide as it is relatively simple and inexpensive to measure, and sufficiently accurate for most adult patients. The Chronic Kidney Disease Epidemiology Collaboration equation based on creatinine (CKD-EPI_Creatinine_) is used worldwide to calculate estimated glomerular filtration rate (eGFR). CKD-EPI_Creatinine_ is recommended by Kidney Disease.

Improving Global Outcome (KDIGO) and the Danish Society of Nephrology for standard diagnosis and treatment of kidney disease [8,9], and it is commonly used for drug prescribing in the clinic. The creatinine-based Cockcroft-Gault equation (CG_Creatinine_) developed in 1973 [10] also continues to be used in pharmacokinetic studies during drug development [11] and sometimes in clinical practice [12,13], despite its lack of performance [14]. However, creatinine is heavily dependent on muscle mass, nutritional status, age, and sex [15,16,17]. Patients undergoing amputation are at particularly high risk of incorrect estimation of kidney function based on creatinine, due to malnutrition and loss of muscle mass [15,16,18,19], which may lead to inappropriate prescribing of renally-eliminated medications.

Cystatin C is another endogenous biomarker, which is less dependent on age, sex and muscle mass compared to creatinine [20] and, therefore, recommended for patients with low muscle mass [8,9]. A combined creatinine and cystatin C equation (CKD-EPI_Combined_) is thought to be the most accurate across a full range of GFRs and, in particular, patient subgroups [9,21,22,23,24], while the equation using cystatin C alone (CKD-EPI_CystatinC_) was as accurate as CKD-EPI_Combined_ in elderly patients with very low BMI [25]. Another study concluded that cystatin C is preferable to creatinine, following traumatic amputation in young soldiers [18]. However, no studies, to our knowledge, have focused on the impact of amputation on eGFR and dose recommendations in non-traumatic amputation patients.

Our objectives in this study were: 1) to assess eGFR before and after amputation based on CKD-EPI_Creatinine_, CKD-EPI_CystatinC_, and CKD-EPI_Combined_, and 2) to estimate whether prescribing recommendations for renally-eliminated medications and the number of inappropriate prescriptions of renally-eliminated medications differ among the different CKD-EPI equations.

## 2. Methods and Materials

### 2.1. Study Design

This is a sub-study to the Time to Eat Study, a longitudinal, single-center cohort study, which aimed to describe and measure the association between dietary intake and inflammation in patients with non-traumatic amputation. The study was approved by The Research Ethics Committees (H-15003299) and the Data Protection Agency (AHH-2014-018 03110).

### 2.2. Settings and Participants

Patients were included at the Orthopaedic Amputation Unit at Copenhagen University Hospital, Hvidovre, Denmark, between May 2015 and November 2016. The indication for amputation among all patients was gangrene, due to poor circulation in the lower extremities. The inclusion criteria were: Above 50 years of age and able to give an informed consent. Exclusion criteria were: Pathologic or traumatic indications, intravenous drug-abuse, major surgical intervention within four weeks before inclusion, amputation later than 14 days after inclusion, toe amputation, and parenteral nutrition use at admission. For the current study, patients were also excluded if they received an ankle amputation, had a diagnosis of acute kidney injury (AKI), or were in dialysis.

### 2.3. Study Data

Serum creatinine and cystatin C were measured at the Clinical Biochemical Department at Copenhagen University Hospital, Hvidovre, Denmark, on a Roche Cobas^®^ c 8000 701/702. Creatinine was measured with a module instrument using the Roche, Basel, Switzerland, Creatinine Plus version 2 *IDMS*-*traceable* enzymatic assay (coefficient of variation 1.5%). Cystatin C was measured using the Roche Cystatin C Tina-quant generation 2 particle-enhanced immunonephelometric assay (coefficient of variation 2.2%). Both methods were standardized and validated in correspondence with Kidney Disease Improving Global Outcomes (KDIGO) guidelines [9]. The neutrophil, gelatinase-associated lipocalin (NGAL) was measured with the NGAL Test^TM^ from Bioporto^®^, Copenhagen, Denmark, on a Roche Cobas^®^, Basel, Switzerland, c 8000 c501/c502 particle-enhanced, turbidimetric immunoassay (coefficient of variation 3.7%). C-reactive protein (CRP) and thyroid stimulating hormone (TSH) were obtained from routine lab tests. TSH was measured on a Roche Cobas^®^, Basel, Switzerland, c8000 e602 competitive electrochemiluminescence immunoassay (coefficient of variation 3.8%). Reference interval for TSH was 0.4–4.8 ng/mL.

Blood samples were obtained at inclusion (day −1), day of operation (day 0), and days 1, 3, 5, and 10 after amputation. Day 0 was defined as the before amputation blood sample. If this sample was missing, then day −1 was used instead. In cases where no blood samples were available on day 5 or 10, the missing sample was replaced by the closest sample taken on day 6, 7, 8, 9, or 11.

The CKD-EPI equations were used to calculate eGFR, and eGFR was classified into one of five categories according to the 2003 National Kidney Foundation Kidney Disease Outcomes Quality Initiative Classification (K/DOQI) [26]. To investigate the complexity of kidney function, the number of patients switching at least one kidney function category was calculated for each time interval.

To investigate the potential cases of undiagnosed AKI, we identified patients with an increase in serum creatinine of ≥26.5 µmol/l within 48 hours or 50% within seven days as per 2012 KDIGO AKI guidelines [19]. Three time intervals were used to determine creatinine change within 48 hours: day 0–1, day 1–3, and day 3–5. Five time intervals were used to determine creatinine change within seven days: day 0–3, day 0–5, day 1–5, day 3–10, and day 5–10.

Weight and BMI were adjusted for a previous amputation at baseline. Muscle strength was estimated by handgrip strength on a hand-held dynamometry device [27,28]. Smoking status and comorbidities that were registered with International Statistical Classification of Diseases and Related Health Problems 10th Revision (ICD-10) codes and obtained from patient records. Smoking status was divided into current, previous, and never-smokers. Amputations were coded both by amputation type and extent of muscle loss. Amputation type was either transtibial (KNGQ19) or transfemoral (KNFQ19), and the extent of muscle loss was grouped as either major or minor muscle loss (Table 1).

Medication information was obtained from the Danish electronic patient medication record and included all medications administered from hospital admission to discharge, or 11 days post-operation, whichever occurred first. Medication prescribing recommendations based on eGFR were obtained from Renbase^®^. Medications were labeled, “renally-eliminated medications” if dose adjustments were recommended at GFR ≤ 90 ml/min/1.73 m^2^. Total daily dose of renally-eliminated medications administered post-operation was compared to dose recommendations in Renbase^®^, and a “dosing discrepancy” was defined as any case in which total daily dose exceeded dose recommendations [3].

### 2.4. Statistical Analysis

CKD-EPI equations, both before, and after amputation were compared using the Friedman’s test. To evaluate changes in eGFR from pre- to post-operation, mixed models were used with patient identification modelled as a random effect, considering repeated measurement across investigation days with a compound symmetry co-variance structure and a binary parameter, indicating whether the measurements were before, or after, surgery as fixed effect. To investigate whether patients with major muscle loss had a larger increase in eGFR than patients with minor muscle loss from pre- to post-operation, an interaction term between the binary surgery parameter (before/after) and muscle loss (minor/major) was included in the model. Finally, a fixed parameter for days after amputation was included in a separate model to investigate whether the eGFR changes after amputation were stable or simply a reflection of post-operative recovery. To investigate the robustness of the findings, statistical tests were repeated after excluding patients with potential AKI.

Goodness of fit for the linear regression models was determined by visual inspection of a histogram of residuals to test normal distribution, and a scatter plot of residuals versus predicted values to test variance homogeneity. If needed, the dependent variable was transformed to improve goodness of fit.

All calculations and statistical analyses were performed in SAS software, SAS Enterprise Guide, Version 7.1. Copyright © SAS Institute Inc., Cary, NC, USA. Figures were created in RStudio 3.2.3., Integrated Development for R. RStudio, Inc., Boston, MA, USA. For all statistical tests, *p* ≤ 0.05 was considered statistically significant.

## 3. Results

Data from a total of 42 patients were included in the Time to Eat Study. Four patients were excluded in the present study due to ankle amputation (*n* = 3) or dialysis (*n* = 1). The median age of participants was 75 years, and 29% were females. Table 2 includes information about baseline characteristics. One patient died prior to day 3, and data from this patient were included when possible. Five patients (13.2%) were identified with potential AKI, but none of these patients were diagnosed with AKI during hospitalization.

Prior to amputation, 30 (81%) patients were within TSH reference interval, whereas all patients had a CRP greater than reference (>10 mg/dL) (Table 2). After amputation, the data from 38 patients (day 1) revealed the median (range) serum creatinine and cystatin C values of 0.9 (0.4–3.8) mg/dL and 1.3 (0.8–3.5) mg/L, whereas data from 37 patients (day 3) revealed values of 0.9 (0.4–2.7) mg/dL and 1.3 (0.7–3.5) mg/L, respectively. After amputation (day 3) the patients had a median (range) CRP of 120 (37–340) mg/dL and all patients had values greater than reference. Approximately half of the patients had a NGAL value greater than 150 ng/mL, both before, and after, amputation with a median (range) value of 159 (53–1053) and 147 (59–695) ng/mL before, and after, amputation, respectively. GFR estimates were significantly different between CKD-EPI_Creatinine_, CKD-EPI_CystatinC_, and CKD-EPI_Combined_ both before, and after, amputation (Friedman’s test: all *p* < 0.01) (Table 3).

Figure 1 shows the individual estimated glomerular filtration rate (eGFR) curves for each patient in the study during hospitalization.

### 3.1. Impact of Amputation on Kidney Function

GFR estimates for all equations increased significantly from pre- to post-operation (Table 3). However, these increases were not significantly different among the three equations. Analyses of interaction terms indicated that patients with major muscle loss had a larger increase in eGFR compared to patients with minor muscle loss (*p* = 0.1 for eGFR_Creatinine_, *p* = 0.02 for eGFR_CystatinC_ and *p* = 0.03 for eGFR_Combined_). However, these differences were no longer significant when patients, with potential AKI (*n* = 5), were excluded (all *p* values ≥ 0.14). Likewise the differences were insignificant (all *p* values ≥ 0.07) if patients with potential AKI were calculated based on the post-operation creatinine measurements only (*n* = 4).

The investigation of the changes after amputation as a reflection of post-operative recovery showed that GFR estimates decreased significantly from post-operative day 5 to post-operative day 10 for both CKD-EPI_CystatinC_ (−6.8 mL/min/1.73m^2^, 95% CI: −9.7; −3.8, *p* < 0.01) and CKD-EPI_Combined_ (−4.5 mL/min/1.73 m^2^, 95% CI: −7.7; −1.3, *p* < 0.01). Similar results were obtained when patients with potential AKI were excluded.

### 3.2. Prescribing Renally-Eliminated Medications

In the post-operative period, patients were prescribed a median (range) of 12 (6–25) medications, including 7 (2–14) renally-eliminated medications. Morphine was the most commonly prescribed medication, followed by oxycodone, dalteparin, and gabapentin. In total, at least one drug was prescribed at higher than recommended dose according to Renbase^®^ in 10.8% of patients (12 prescriptions) according to eGFR_Creatinine_, 37.8% of patients (37 prescriptions) according to eGFR_CystatinC_, or 17.6% of patients (16 prescriptions) according to eGFR_Combined_. When switching from CKD-EPI_Creatinine_ to CKD-EPI_CystatinC_, prescribing recommendations were particularly affected for morphine, gabapentin, and metformin (Table 4). A similar pattern was observed when patients with potential AKI were excluded. Notably, however, potential AKI was identified in 3 of 14 patients with dosing discrepancies according to eGFR_Creatinine_ or eGFR_Combined_, and 4 of 14 patients with dosing discrepancies according to eGFR_CystatinC_.

### 3.3. Kidney Function Estimates During Hospitalization

Table 5 shows the number of patients whose kidney function classification changed during hospitalization. With CKD-EPI_Creatinine_, 6.1% to 16.2% of patients switched classification between days.

## 4. Discussion

In a cohort of amputation patients, we found that eGFR differs significantly between CKD-EPI_Creatinine_, CKD-EPI_CystatinC_, and CKD-EPI_Combined_ both before, and after, amputation. These differences may affect how patients are classified according to kidney function and, therefore, influence the prescribing of renally-eliminated medications. We also found that eGFR increases significantly after amputation for all CKD-EPI equations. However, we found no significant differences between CKD-EPI equations in the degree of this effect. Finally, we identified fluctuations in GFR classification during hospitalization, highlighting the challenge in prescribing renally-eliminated medications. These findings stress the importance of further work to determine how to most accurately estimate kidney function in this frail and vulnerable patient population.

### 4.1. eGFR Differences Before, and After, Amputation

The observed differences between creatinine- and cystatin C-based equations both before, and after, amputation are two to four-fold larger in our population compared to studies in older hospitalized patients without amputation [3,30,31,32]. The differences in our study are about the same size as reported in the Northern Manhattan Study of 2,988 elderly patients with diverse racial and ethnic backgrounds. Patients in this study had a mean age of 69.2 years, and 21% of patients had been diagnosed with diabetes, 73% with hypertension and 24% with cardiac disease [33]. Patients in our study are characterized by older age, decreased muscle mass, high inflammation, morbidity, fragility, and post-operative stress [34,35]. Creatinine is dependent on muscle mass, sex, and age [20,36,37], while cystatin C has been suggested to increase with inflammation [38] and other extra-renal factors such as smoking, thyroid dysfunction, glucocorticoid use [3,36,37], and perhaps cardiovascular diseases [39]. We have previously shown in elderly medical patients that age > 80 years, low handgrip strength (<26 kg for males and <16 kg for females), CRP > 10 mg/dL, NGAL > 150 ng/mL, and current smoking are significantly associated with differences in GFR estimates based on creatinine and cystatin C [3]. These features are also common in our patient cohort, which could explain the significant differences in eGFR based on creatinine and cystatin C before amputation. Another challenge with using CKD-EPI_Creatinine_ in elderly patients is that the equation was developed in a patient cohort where only 4% (*n* = 217) of participants were over 70 years old [40]. We expect that the high age, low muscle mass, and inflammation in particular are responsible for the observed eGFR differences between equations before amputation.

Thurlow et al. reported that eGFR_Creatinine_ is influenced by amputation in young male soldiers with a traumatic injury. The authors found that amputation led to an average eGFR increase of 20 mL/min/1.73 m^2^ for eGFR_Creatinine_, but only 11 mL/min/1.73 m^2^ for eGFR_Combined_ and no significant change for eGFR_CystatinC_ [18]. We did not find that amputation affected eGFR_Creatinine_ more than eGFR_Combined_ or eGFR_CystatinC_. This result is likely due to the nature of our patient population. First, low muscle mass before amputation is a plausible explanation for the smaller increase in eGFR_Creatinine_. This effect is supported by the significant differences between equations prior to amputation. Serum creatinine and cystatin C levels are also affected by non-renal factors, and it is possible that creatinine or cystatin C did not reach steady state during our study period. Altogether, these causes could have masked the effect of amputation on eGFR_CystatinC_, but it is not possible to investigate the influence of these factors without directly measuring kidney function. Finally, our results suggest that estimates of kidney function in elderly non-traumatic amputation patients are highly complex regardless of which biomarker is used.

### 4.2. Choice of eGFR Equation and Biomarker to Amputation Patients

While our study cannot conclude anything about the accuracy of eGFR equations, it is clear that the choice of equation has a major impact on prescribing recommendations for renally-eliminated medications in our group of patients. More work is needed to determine the most accurate eGFR equation for amputation patients, but it is unlikely that this equation will rely only on creatinine. The CG_Creatinine_ equation continues to be used in pharmacokinetic studies during drug development [11]. Due to the lack of performance of CG_Creatinine_ [14], the alternative CKD-EPI_Creatinine_ equation is sometimes used for drug prescribing in the clinic [41], and often preferred for the diagnosis and treatment of kidney disease [19]. CG_Creatinine_ is also impractical because it requires continuous weight measurements. Weight measurements are typically not recorded in amputation patients because they are difficult to obtain if the patient is immobilized, and are typically inaccurate due to unstable fluid and food intake, edema, and blood loss. However, a weight measurement is necessary to determine body surface area, which is used to convert eGFR based on CKD-EPI from normalized units (mL/min/1.73 m^2^) to absolute units (mL/min). For drug development, the European Medicines Agency only recommends eGFR in absolute values [42], while the US Food and Drug Administration accept both normalized and absolute values [11]. The choice of eGFR unit can impact prescribing recommendations—particularly in patients with small or large body surface area. Therefore, future equations should rely less heavily on weight measurements, or utilize biomarkers which are less dependent on weight. Ideally, these equations should be built on a panel of kidney biomarkers and account for individual patient characteristics [37,43]. Alternatively, gold standard markers could be made more accessible in clinical settings. This can be done by introducing single-plasma samples [44] or dried blood spot testing [45], which should be considered as alternatives to eGFR.

### 4.3. Choice of eGFR Biomarkers Impact on Dosing Discrepancies

Our results suggest that switching between eGFR_Creatinine_, eGFR_CystatinC_, and eGFR_Combined_ would lead to different dosing recommendations for renally-eliminated medications, and this pattern remained after patients with potential AKI were excluded. Dosing discrepancies occurred in 10.8% to 37.8% of patients depending on the choice of the eGFR equation. As expected, eGFR_CystatinC_ yielded the lowest eGFR and therefore the highest number of dosing discrepancies. Recent studies have similarly identified that GFR estimates are lower with cystatin C-based equations compared to creatinine-based equations in hospitalized patients without amputation and these differences had direct influences on prescribing recommendations for renally-eliminated medications [3,30,31]. However, none of these studies investigated the impact in patients undergoing amputation, making this study unique. The rate of dosing discrepancies for CKD-EPI_Creatinine_ in our study population is high, but in concordance with existing literature regarding prescribing practices in patients with impaired kidney function [4,46,47,48,49]. However, the observed increase in eGFR when switching to CKD-EPI_CystatinC_ had a greater impact on dosing discrepancies compared to other studies of kidney function estimates in elderly patients [3].

Given the high number of renally-eliminated medications prescribed to amputation patients, the risk of adverse drug effects due to dosing discrepancies reinforces the importance of obtaining accurate kidney function estimates. Morphine was the most commonly prescribed renally-eliminated medication in our cohort, and overdosing of morphine increases the risk of dangerous side effects, including respiratory depression [50]. In a previous study, we also identified significant differences in dosing of analgesics to the elderly using different eGFR_Creatinine_ equations [51]. Dabigatran, gabapentin, and metformin are eliminated almost entirely by the kidneys in an unchanged form, and overdosing of these medications in frail multi-morbid patients may lead to life-threatening adverse drug reactions. However, these medications are often necessary for amputation patients and are required for post-operative recovery. In cases where an accurate estimate of kidney function cannot be obtained, we suggest switching to drugs, which are less dependent on renal function. In the case of pain management, for example, fentanyl or another opioid analogue could be considered in place of morphine.

The major differences between eGFR biomarkers in our study population also raise important questions about the design of pharmacokinetic studies during drug development. We believe that prescribing guidelines should be based on large pharmacokinetic characterization studies that use a gold standard marker, as well as a panel of eGFR biomarkers with a wide range of demographic representation. Data from these studies should be accessible for clinicians and displayed on medication package inserts [3,13]. From a regulatory perspective, medication prescribers should use the eGFR equation indicated in the summary of product characteristics (SPC). This equation is CG_Creatinine_ for the most currently marketed drugs [3,12]. However, clinicians should also know the limitations of particular eGFR equations and recognize that pharmacokinetic characterization studies do not always reflect what is relevant for the individual patient.

Finally, we observed clinically relevant fluctuations in GFR classification among amputation patients during hospitalization. This finding highlights the challenge that clinicians face when prescribing renally-eliminated medications to patients with unstable eGFR. While we have discussed the risks of drug overdoses, drug under-dosing could lead to insufficient treatment of post-operative pain, infection, and thrombosis. For these reasons, we recommend that clinicians are particularly attentive to prescribing renally-eliminated medications in amputation patients and apply therapeutic drug monitoring whenever possible. They should also consider whether patients should have their kidney function measured with a gold standard until further evidence about the accuracy of GFR estimates for amputation patients is available.

### 4.4. Strengths and Limitations

The primary strengths of this study were the availability of repeated measurements, which made it possible to study the changes in kidney function over time, and the access to reliable medication records. Generally, this study reflects a daily clinical problem in the surgery department and has direct implications for clinical practice in amputation patients. However, we do not have creatinine measurements after 10 days post-operation, so we are unable to determine whether creatinine or cystatin C reach a steady-state during the study period. Our study also references dosing guidelines from Renbase^®^, which is commonly used by Danish clinicians through pro.medicin.dk [6]. Renbase^®^ has also been referenced in several Northern European studies [3,4,47,51]. It contains prescribing recommendations for more medications than other databases and is believed to reflect daily clinical practice in Denmark. However, access to Renbase^®^ requires a payment and is, therefore, not available to all clinicians worldwide. Access to high quality databases is a well-known challenge and a limitation for the interpretation of our results. Another limitation to our study is the lack of gold standard markers for measuring kidney function, making it impossible to determine the absolute accuracy of CKD-EPI equations. Finally, no urine samples were available for this study, so we could not evaluate the third KDIGO criterion for assessing AKI. We only identified one patient with potential AKI based on a change in creatinine from pre- to post-operation, and excluding this patient did not influence our results. However, comparing creatinine measurements across amputation is not a sensitive determinant of AKI, due to the loss of muscle mass.

## 5. Conclusions

Non-traumatic amputation of a lower extremity has a significant effect on eGFR based on creatinine, cystatin C, or both. We found significant differences between eGFR from CKD-EPI_Creatinine_, CKD-EPI_CystatinC_, and CKD-EPI_Combined_, both before, and after, amputation, and these differences impact how renally-eliminated medications are prescribed. We also identified unstable eGFR classification during hospitalization, thereby highlighting the risk of over- or under-dosing renally-eliminated medications and highlighting the need to further study how kidney function is most accurately determined in this vulnerable patient group.

## Figures and Tables

**Figure 1 jcm-08-00089-f001:**
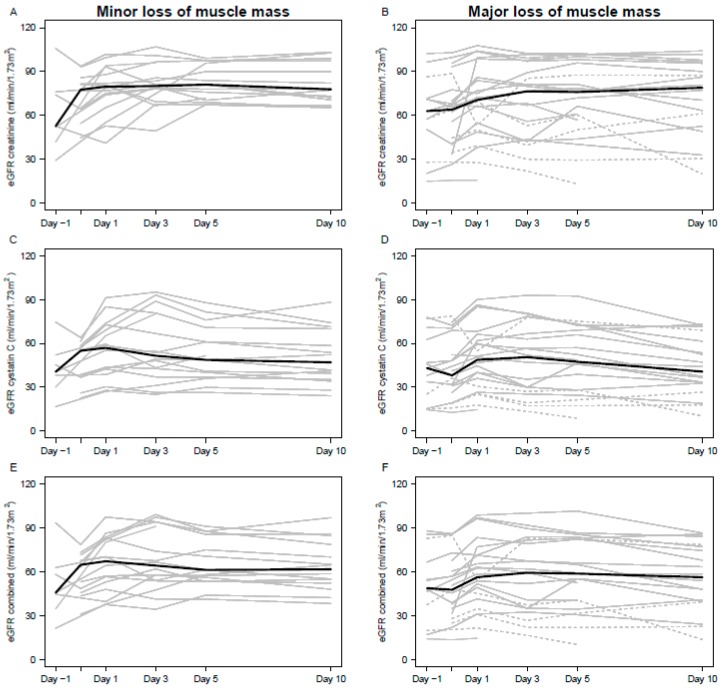
The level of eGFR during lower extremity amputation (*n* = 38). Light grey lines represent each patient of which dashed lines are patients with potential acute kidney injury. Solid, black lines are the medians.

**Table 1 jcm-08-00089-t001:** Definition of minor and major muscle loss according to the amputation procedure.

Minor Muscle Loss	Major Muscle Loss
Amputation at or below knee or Transfemoral amputation with a prior transtibial amputation on the same leg	Transfemoral amputation with no prior amputations on the same leg

**Table 2 jcm-08-00089-t002:** Patient demographics.

Characteristics	All Patients (*n* = 38)	Patients with a Minor Loss of Muscle (*n* = 22)	Patients with a Major Loss of Muscle (*n* = 16)
Age, years, median (range)	75 (53–95)	74 (53–89)	78 (60–95)
Age ≥ 80 years, *n* (%)	14 (37)	6 (27)	8 (50)
Sex, female, *n* (%)	11 (29)	4 (18)	7 (44)
Weight, kg, median (range)	67 (43–112)	77 (43–112)	60 (46–92)
BMI ^a^, kg/m^2^, median (range)	22 (15–34)	23 (15–34)	22 (17–31)
BMI ≤ 18.5 kg/m^2^, *n* (%)	5 (13)	2 (9)	3 (19)
Kidney function ^b^			
Serum creatinine, mg/dL, median (range)	1.1 (0.5–3.8)	1.1 (0.5–3.8)	0.9 (0.6–2.1)
Serum cystatin C, mg/L, median (range)	1.5 (0.9–3.8)	1.5 (0.9–3.8)	1.4 (1.1–3.0)
Smoking, current, *n* (%)	17 (45)	11 (50)	6 (38)
TSH, ng/mL, median (range) ^c^	1.7 (0.1–26.3)	1.7 (0.4–8.8)	1.7 (0–26.3)
CRP ^d^, mg/dL, median (range)	88 (14–240)	68 (14–240)	98 (23–220)
Handgrip strength ^e^, kg, median (range)	22 (4–40)	24 (9–38)	18 (4–40)
Low handgrip strength ^f^, *n* (%)	21 (57)	11 (52)	10 (63)
Comorbidities			
Diabetes, *n* (%)	17 (45)	10 (45)	7 (44)
Atherosclerosis, *n* (%)	29 (76)	15 (68)	14 (88)
Hypertension, *n* (%)	22 (58)	13 (59)	9 (56)

a: Missing values for three patients. If missing, then day 2 or 4, post-operation was used. b: All values read from day of operation (day 0). If missing, values read from inclusion (day −1). c: Missing value for one patient. d: Missing values for six patients. e: Missing value for one patient. All values read from day of operation (day 0). If missing, values read from inclusion (day −1). f: Low handgrip strength was defined as < 26 kg for males and <16 kg for females [29]. BMI: Body Mass Index. TSH: Thyroid stimulating hormone. CRP: C-reactive protein.

**Table 3 jcm-08-00089-t003:** Estimated glomerular filtration rate (eGFR) from the Chronic Kidney Disease Epidemiology Collaboration (CKD-EPI) before, and after, amputation, and mean difference between equations.

	Raw Data		Mixed Models	
	Before, Median (Range)	After, Median (Range)	Mean Difference, (95% CI)	*p* ^b^
eGFR_Creatinine_, mL/min/1.73 m^2^	65 (15–103)	80 (22–107)	8.5 (5.1; 11.8)	< 0.01
eGFR_CystatinC_ mL/min/1.73 m^2^	38 (13–79)	51 (13–95)	6.1 (3.6; 8.6)	< 0.01
eGFR_Combined_ mL/min/1.73 m^2^	48 (13–86)	62 (16–100)	7.4 (4.7; 10)	< 0.01
Friedmans test, *p* ^a^	< 0.01	< 0.01		

a: Friedman’s test for the differences among eGFR based on creatinine, cystatin C, and the combination before and after amputation, respectively. b: Mixed model test for the difference between before, and after, amputation for each CKD-EPI equation, respectively.

**Table 4 jcm-08-00089-t004:** Number of patients with dosing discrepancies during hospitalization according to eGFR on day 3 post-operation using CKD-EPI equations.

Active Substance	Patients with Dosing Discrepancies, *n* (%)			Total Patients Prescribed, *n* (%)
	CKD-EPI equation			
	Creatinine	Cystatin C	Combined	
Morphine (N02AA01)	2 (8)	6 (25)	3 (13)	24 (65)
Gabapentin (N03AX12)	3 (15)	7 (35)	4 (20)	20 (54)
Simvastatin (C10AA01)	0 (0)	1 (7)	0 (0)	14 (38)
Zopiclone (N05CF01)	1 (9)	2 (18)	1 (9)	11 (30)
Metformin (A10BA02)	2 (29)	5 (71)	3 (43)	7 (19)
Allopurinol (M04AA01)	0 (0)	1 (33)	0 (0)	3 (8)
Hydrochlorothiazide (C03AA03)	0 (0)	1 (33)	0 (0)	3 (8)
Mirtazapine (N06AX11)	0 (0)	1 (33)	0 (0)	3 (8)
Sitagliptin (A10BH01)	1 (33)	3 (100)	1 (33)	3 (8)
Bendroflumethiazide (C03AB01)	0 (0)	1 (50)	0 (0)	2 (5)
Cetirizine (R06AE07)	1 (50)	1 (50)	1 (50)	2 (5)
Ciprofloxacin (J01MA02)	0 (0)	0 (0)	0 (0)	2 (5)
Dabigatran (B01AE07)	0 (0)	1 (50)	0 (0)	2 (5)
Magnesium (A02AA04)	0 (0)	1 (50)	1 (50)	2 (5)
Metoclopramide (A03FA01)	0 (0)	1 (50)	0 (0)	2 (5)
Colchicin (M04AC01)	0 (0)	1 (100)	0 (0)	1 (3)
Venlafaxine (N06AX16)	1 (100)	1 (100)	1 (100)	1 (3)
Total, patients	4 (11)	14 (39)	6 (17)	-
Total, patients without potential AKI (*n* = 31)	1 (3)	10 (28)	3 (8)	-

**Table 5 jcm-08-00089-t005:** Patients who are re-classified (change or decrease) at least one category. Change refers to both patients who are re-classified into a higher or lower category, whereas decrease refers to patients who are re-classified into a lower category.

	Day −1–Day 0	Day 0–Day 1	Day 1–Day 3	Day 3–Day 5	Day 5–Day 10
eGFR_Creatinine_, change, *n* (%) decrease, *n* (%)	2 (6.1)	5 (13.2)	6 (16.2)	6 (16.2)	4 (12.5)
	0 (0.0)	1 (2.6)	3 (8.1)	2 (5.6)	2 (6.3)
eGFR_CystatinC_, change, *n* (%) decrease, *n* (%)	2 (6.1)	8 (21.1)	7 (18.9)	3 (8.3)	6 (18.8)
	0 (0.0)	1 (2.6)	5 (13.5)	1 (2.8)	5 (15.6)
eGFR_Combined_, change, *n* (%) decrease, *n* (%)	0 (0.0)	7 (18.4)	6 (16.2)	5 (13.9)	5 (15.6)
	0 (0.0)	1 (2.6)	4 (10.8)	3 (8.3)	4 (12.5)

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
