# Peer review of "Differences in Kidney Function Estimates Based on Creatinine and/or Cystatin C in Non-Traumatic Amputation Patients and Their Impact on Drug Prescribing"

_jcm, 2019, doi:10.3390/jcm8010089_

Reviewer 1 Report

The authors have improved their manuscript by clarifying the ambiguities noted in the previous version. There is one remaining point. The authors have made revisions to the manuscript about the models used to describe the relationships between eGFR equations/NGAL e.g. section 4.2 of Discussion. However, the question remains: what is the biological justification for fitting the data using polynomial curves for these models? Alternative examples of curves that could have been used include linear and exponential curves – why were these not used instead? The reason for raising this point is that fitting a curve to data is not just about maximising the R squared – it should be to come up with a relationship that is biologically plausible. Currently, at least some of the curves are not biologically plausible. The eGFR combined vs NGAL curve is a second order polynomial that means that eGFR combined is high at very high NGAL values (e.g. 1000 ng/mL). This is hard to understand. Also hard to understand is that eGFR creatinine is low at very high eGFR cystatin C values (e.g. 200 mL/min/1.73m2). Have other investigators used these models? Or were these models arbitrarily selected, without considering biological plausibility? If the selection was arbitrary, this should be stated, and the biological implausibility in the relationships noted as a limitation of these models (as described above).

Reviewer 2 Report

Thank you very much for the opportunity to review the article by Aakjaer, et al. entitled, “Differences in kidney function estimates based on creatinine and/or cystatin C in non-traumatic amputation patients and their impact on drug prescribing.”  This was a secondary analysis of a prospectively collected cohort designed to explore the impact of nontraumatic amputation on perceived kidney function and corresponding drug doses.  This is one of the first studies looking at a nontraumatic amputee population and the study had rigorous sampling.  There are several recommendations I would make that would improve the manuscript.

Major feedback:

The manuscript feels approximately 25% too long.  Would recommend making the introduction more succinct and removing the extra historical data.  Would recommend removing much of the first paragraph of 4.1. in the discussion as it refers to different questions in non-amputees and re-evaluate the number of references, to target ones most closely aligned with this study question.

The NGAL piece should be removed.  NGAL is not a kidney “function” biomarker, rather an upregulated protein expressed in the context of AKI (and other disease states such as inflammation as the authors point out), most commonly from an ischemic event.  While in addition to AKI, the authors make reference to associations between NGAL and long-term risk of CKD, that wasn’t measured or evaluated in this study.  Including it in a study about eGFR which is the primary determinant of drug clearance is discordant from the overall objective and distracting.  It seems the authors to some degree feel similarly as the summative aspect of their results on page 9, includes no mention of NGAL. 

To assess AKI, the authors look at KDIGO guideline based increase in SCr during their clinical course.  By definition in a patient who recently underwent an amputation, especially a major one, this definition will be insensitive for detection of AKI.  In fact, a patient before and after amputation, especially with major muscle loss, with a consistent SCr would assuredly have experienced occult injury as one would expect SCr to go down which has been demonstrated in other studies.  For this reason then, it’s not only impossible to draw conclusions about incidence and severity of AKI, but also understand the performance of NGAL as a predictor.  I think the authors tried to address this by using the word “possible AKI,” but this remains insufficient in my opinion.  Post-amputation baseline could be re-established and renal trends thereafter for AKI incidence described, but it’s not evident that this baseline was used, and would be reluctant to refer to this as AKI.

Other studies have recently been published on this topic which decreases the novelty.  These should be referenced (PMID: 30120844, PMID: 30372593, PMID: 29855865), but it should be highlighted that they did not focus on before/after amputation which makes this study unique.

Minor feedback:

Introduction:

The statement on page 2, line 74 re: the combined equation: this equation is most accurate across the spectrum of kidney function, indeed especially at the high end of kidney function where it exhibits better performance than MDRD and others.

Results

Raw SCr and SCysC data should be provided at least at Day -1, D0, D+1

Recommend remove figure 2, the relationship of these equations with each other has been well characterized in the literature in much large samples with measured GFR and the renal trends surrounding amputation are captured in other places in the paper.  

The first sentence in 3.1 is redundant with the preceding aspect of the results

Discussion

The way the first sentence of 4.1. is written it makes it seem that you’re referring to community-dwelling populations with amputation. 

Page 9 line 283, it’s not just that they did not reach steady state, but that their observed serum concentrations are fundamentally affected by both renal and non-renal determinants which, without measured GFR, could not be separated.

Recommend soften the statement made on pg 10, line 300-302, CKD-EPI has not broadly replaced CGcreatinine.  It is an alternative, but not necessarily a recommended one.  

Recommend adjusting the comment about weight in 4.2, line 304-306.  Drug dosing with CKD EPI should use mL/min, not mL/min/1.73m2 (unless maybe with chemotherapy).  In order to re-express the unit, height and weight are used to calculate BSA, therefore weight would also be necessary to effectively use CKD EPI in drug dosing.

It should also be strongly emphasized that application of new eGFR tools to historical dosing thresholds is ill-advised as the authors rightly demonstrate that dose recommendations change and often decrease when cystatin C is included.  Without corresponding pharmacokinetic data, one cannot know what the clinical ramifications of this choice are, and drug specific dosing models with newer eGFR tools need to be developed and explicitly described in package inserts and tertiary references.

Overall delivery

There are numerous spelling and grammatical mistakes throughout the manuscript.  As an example Page 2, line 68, “Patients undergoing are at particularly high risk…”

Rather than using “renal risk” drugs, which broadly refers to medications that include renally-eliminated AND nephrotoxic medications, would refer to them as renally-eliminated medications since the study is predicated on drug clearance and exhaustive list of nephrotoxins was not the focus.

Author Response

Round  2

Reviewer 2 Report

I appreciate that the authors took the time to specifically addressed the feedback provided. I have no further concerns and congratulate them on a nice study.